# Active Pairwise Constraint Learning in Constrained Time-Series Clustering for Crop Mapping from Airborne SAR Imagery

Xingli Qin [1,2], Lingli Zhao [3,*], Jie Yang [2], Pingxiang Li [2], Bingfang Wu [1,4], Kaimin Sun [2] and Yubin Xu [5]

1   State Key Laboratory of Remote Sensing Science, Aerospace Information Research Institute, Chinese Academy of Sciences, Beijing 100101, China
2   State Key Laboratory of Information Engineering in Surveying, Mapping and Remote Sensing, Wuhan University, Wuhan 430079, China
3   School of Remote Sensing and Information Engineering, Wuhan University, Wuhan 430079, China
4   College of Resources and Environment, University of Chinese Academy of Sciences, Beijing 100049, China
5   China Academy of Civil Aviation Science and Technology, Beijing 100028, China
*   Correspondence: zhaolingli@whu.edu.cn; Tel.: +86-159-7204-6289

**Abstract:** Airborne SAR is an important data source for crop mapping and has important applications in agricultural monitoring and food safety. However, the incidence-angle effects of airborne SAR imagery decrease the crop mapping accuracy. An active pairwise constraint learning method (APCL) is proposed for constrained time-series clustering to address this problem. APCL constructs two types of instance-level pairwise constraints based on the incidence angles of the samples and a non-iterative batch-mode active selection scheme: the must-link constraint, which links two objects of the same crop type with large differences in backscattering coefficients and the shapes of time-series curves; the cannot-link constraint, which links two objects of different crop types with only small differences in the values of backscattering coefficients. Experiments were conducted using 12 time-series images with incidence angles ranging from 21.2° to 64.3°, and the experimental results prove the effectiveness of APCL in improving crop mapping accuracy. More specifically, when using dynamic time warping (DTW) as the similarity measure, the kappa coefficient obtained by APCL was increased by 9.5%, 8.7%, and 5.2% compared to the results of the three other methods. It provides a new solution for reducing the incidence-angle effects in the crop mapping of airborne SAR time-series images.

**Keywords:** synthetic aperture radar (SAR); crop mapping; time-series images; constrained clustering; active constraint learning

## 1. Introduction

Food security is the foundation and guarantee of human health and social stability. Timely and accurate crop mapping can assist with crop production management, crop yield prediction, crop disaster warning, and crop planting planning, which is of considerable importance to ensure food security [1–3]. Remote sensing technology has become an important means of crop mapping due to its wide coverage, low cost, near real-time performance, and the large amount of information [1,4–6].

Among the numerous earth observation sensors, synthetic aperture radar (SAR) systems on airborne platforms have been gaining popularity in crop mapping applications. On the one hand, the airborne platforms have the advantages of good flexibility and reusability and can guarantee regular and high-frequency observations throughout the growing season, which increases their suitability for agricultural monitoring. On the other hand, as an activate microwave sensor, SAR provides unique information for crop mapping: the microwave signal is sensitive to the dielectric and geometrical properties of crops [7], and the acquired backscattering coefficients are synthesized information of the land cover type, terrain slope, surface roughness, local incidence angle, and dielectric constant [8].

Thus, rich polarimetric features can be extracted by various polarimetric decomposition algorithms [9–13].

The accuracy of crop mapping in a single image is affected by the phenomenon of "same object with different spectra" and "different objects with the same spectra." Several researchers currently use time-series analysis technology to map crops from time-series images based on the differences in the growth trends of the crops over the whole growth period. These methods can be divided into two main categories: supervised and unsupervised methods. In the supervised methods, many researchers have focused on finding remarkably effective features and classifiers to improve the accuracy of specific crop mapping tasks [14–20]. This kind of method can achieve high accuracy but usually requires a large number of training samples, and the acquisition of high-quality training samples can be remarkably time-consuming. Considering the difficulty of the collection of high-quality training samples, the unsupervised crop mapping approaches have attracted increasing attention in recent years. In the unsupervised approaches, some of the methods use crop phenology to establish rules for distinguishing the different crops [21,22]. However, the establishment of rules is highly dependent on expert knowledge and specific crops, which leads to weak universality. Therefore, other methods have been developed using time-series clustering technology [23–28]. Time-series clustering is a process of combining homogeneous time-series data based on a certain similarity measure [29], and its accuracy is mainly affected by the similarity measure criterion, the time-series data representation method, and the clustering method [29]. Time-series clustering technology can be used to cluster the time-series curves of crops effectively due to the differences in the growth cycles and trends of different crops.

However, when applied to the airborne SAR imagery, the crop mapping accuracy may be reduced by the large range of incidence angles. More specifically, when the incidence angles of the images substantially vary, the backscattering coefficients of the same type of crops at the near and far ranges of the image might be quite different, complicating the clustering of these crops into a single group. For the incidence angle effects, some studies [24,30,31] removed the regions with excessively large incidence angles from the image, which cannot fully exploit the image data. Other studies use incidence-angle correction techniques to address these problems, such as normalizing the backscattering coefficient with respect to the incidence angle [32], model and correct for the effect of variation in incidence angle in azimuth line [33], a cosine correction estimated through linear regression [34], and a histogram matching procedure, where the adjustment was based on the lowest two central moments, the mean and variance [35]. However, these techniques are usually suitable for specific data with a small range of incidence angles or specific types of ground objects.

Constrained clustering methods are introduced in this paper to solve the incidence angle effects. Constrained clustering (which is also known as semi-supervised clustering) can introduce background knowledge (also known as side information) to guide a clustering algorithm [36]. The constrained clustering methods focus on the most commonly used K-means based algorithms, in which the clustering algorithm or the objective function is modified such that user constraints are used to guide the algorithm for remarkably appropriate data partitioning [36]. In addition, the current study focused on the most widely used instance-level pairwise constraints [37]: must-link constraints (MLCs) and cannot-link constraints (CLCs). An MLC ensures that two instances should remain in the same cluster, while a CLC ensures that two instances should be in two different clusters [38]. The quality of the constraints plays a critical part in constrained clustering. However, the commonly used constraint generation methods in many studies [36,39–43] are random sampling and manual selection. In random sampling, the constraints are generated by randomly taking pairs of points and generating an MLC or CLC depending on whether they belong to the same class or not. Consequently, the quality of the constraints cannot be guaranteed. In the manual selection, the speckle noise inherent in SAR images will aggravate the difficulty of the identification of constraints. Some studies [44–47] have

attempted to introduce remarkably effective constraint information, but their methods are only applicable in specific domains.

Therefore, an active pairwise constraint learning method (APCL) is introduced to achieve reliable crop mapping from airborne SAR images via constrained time-series clustering. The proposed method integrates the batch-mode active selection, the information contained in the incidence angle, and the characteristics of the crop time-series curves to learn informative instance-level constraints. The method was designed for SAR images with a wide range of incidence angles and was verified in the experiments conducted in this study, where crops from the uninhabited aerial vehicle synthetic aperture radar (UAVSAR) images were mapped using several constrained clustering methods and time-series similarity measures.

## 2. Study Area and Experimental Data

The study area located in the southwest of Winnipeg, Manitoba, Canada, is covered by various crops, as shown in Figure 1a,b. Four types of crops were considered in the crop mapping experiments: oat, corn, canola, and soybean. The reference map for the crop distribution is shown in Figure 1c, which was established in accordance with the land-cover classification map produced by the NASA National Snow and Ice Data Center Distributed Active Archive Center (NSIDC DAAC) [48].

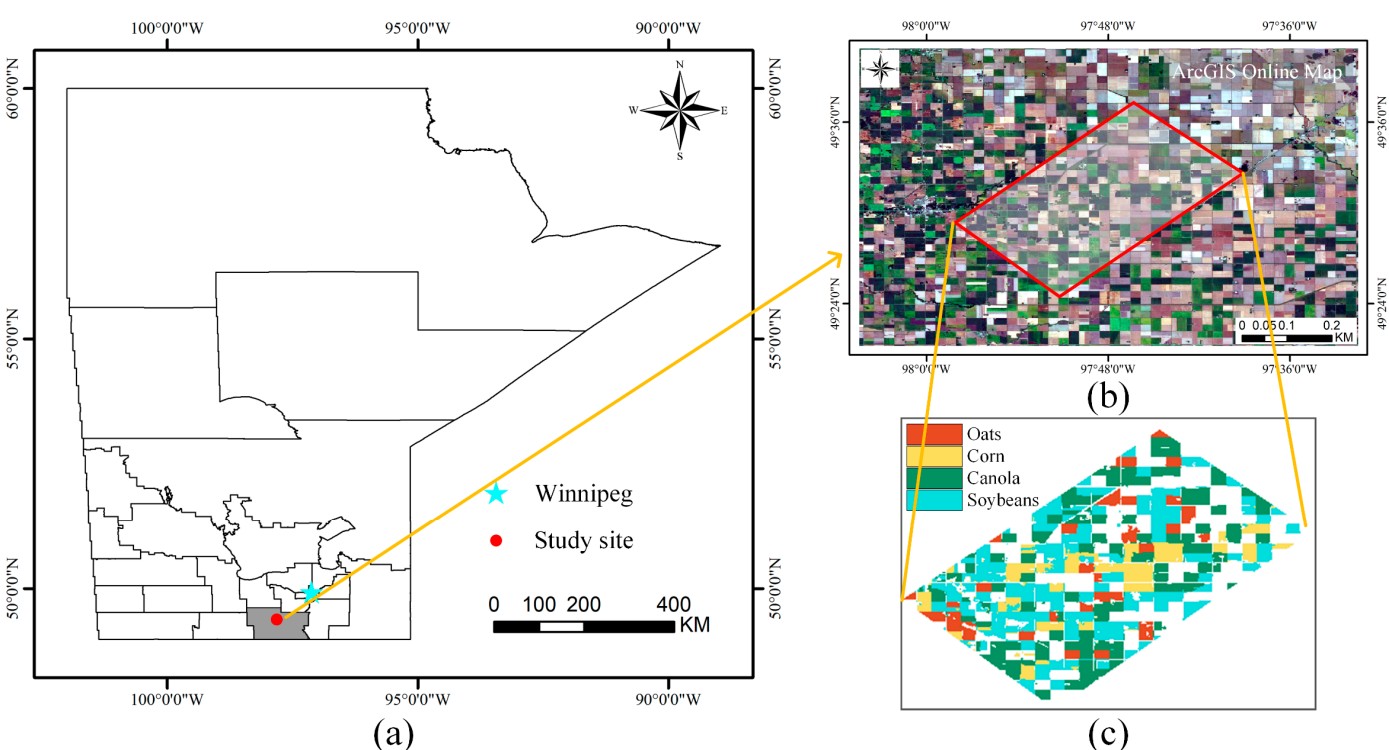

**Figure 1.** Overview of the study area: (**a**) location of the study area; (**b**) overview of the study area in ArcGIS Online Map; (**c**) reference map of crop distribution.

The data used in this study were the UAVSAR images obtained during the Soil Moisture Active Passive Validation Experiment 2012 (SMAPVEX2012). The SMAPVEX12 campaign lasted 43 days (7 June–19 July 2012), and the soil moisture and vegetation conditions changed significantly during the time period. A total of 13 days of UAVSAR data were acquired by the end of the campaign [30]. Twelve UAVSAR images of the same flight line were used for crop mapping. The images were ground-range projected and multi-looked data (GRD) whose preprocessing procedures include radiometric calibration, geometric correction (WGS-84), and multi-look processing (15 pixels in range and 60 pixels in azimuth directions). These images were resampled from $1134 \times 795$ (pixels) to

$378 \times 265$ (pixels) to reduce the time spent in the experiment. The images are multi-looked; thus, any speckle-filtering process was not conducted on the images. A description of the data is provided in Table 1, and the corresponding Pauli RGB images are shown in Figure 2. The local incidence angles of the pixels in the images are shown in Figure 3, where the minimum value is 21.2° (near range) and the maximum value is 64.3° (far range). The local incidence angle is defined as the angle between the radar line-of-sight and the surface normal vector [33,49,50]. The terrain of the study area can be characterized as flat; thus, the incidence angle and local incidence angle are assumed to be the same in this paper. The numbers of pixels for each crop type are presented in Table 2.

**Table 1.** Description of the UAVSAR time-series images.

| Sensor | Look Direction | Band (Frequency) | Incidence Angle (°) | Image Size (Pixels) | Observation Date (2012) |
|---|---|---|---|---|---|
| UAVSAR | Left | L (1.26 GHz) | 21.2–64.3 | $378 \times 265$ | 17, 19, 22, 23, 25, 27, 29 June 5, 10, 13, 14, 17 July |

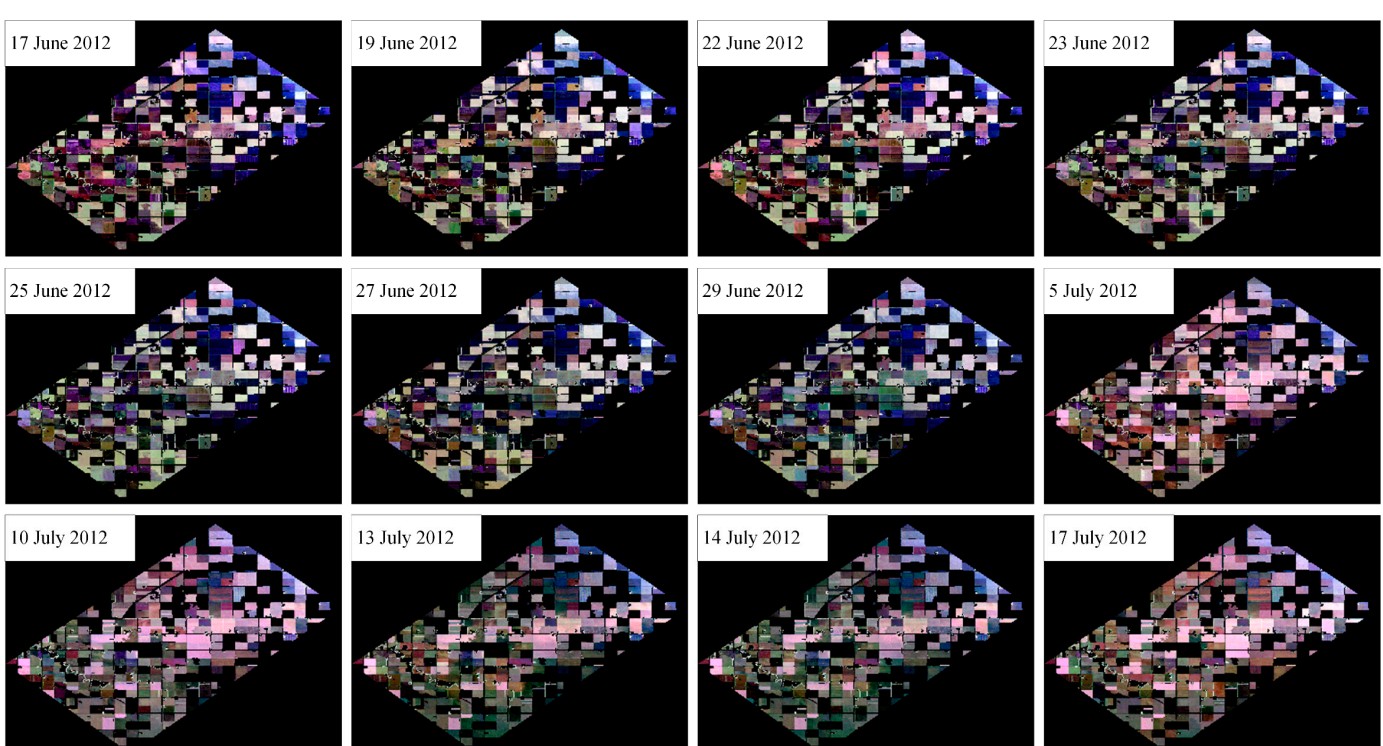

**Figure 2.** Pauli RGB images of the UAVSAR time-series images, with red $|S_{HH} - S_{VV}|^2$, green $4|S_{HV}|^2$, and blue $|S_{HH} + S_{VV}|^2$.

**Table 2.** Number of pixels of each crop.

| Crop Type | Number of Pixels |
|---|---|
| Oats | 3454 |
| Corn | 4395 |
| Canola | 10,426 |
| Soybean | 12,330 |
| Total | 30,605 |

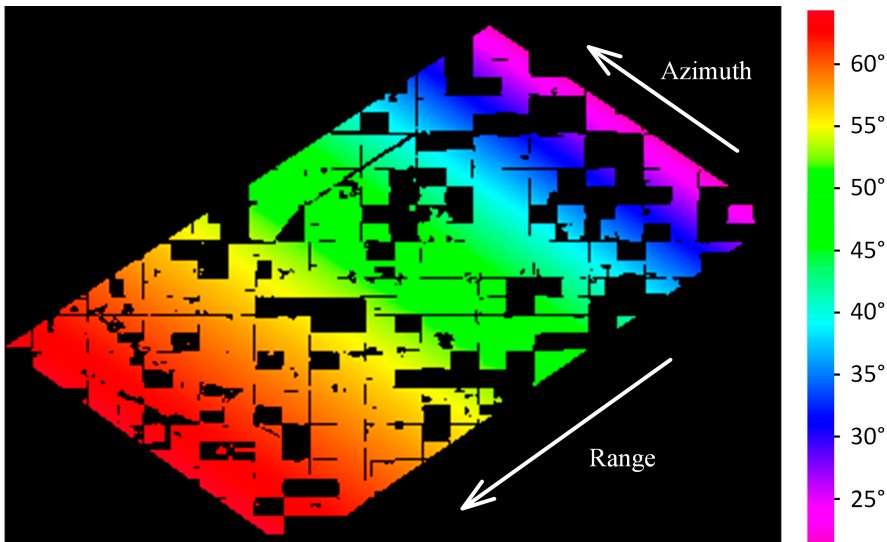

**Figure 3.** Local incidence angles of the samples.

The Pauli RGB images reveal that the backscattering characteristics of the same crop at the near and far ranges are quite different. The backscattering coefficients are mainly affected by the terrain slope, soil moisture, geometric structure, and incidence angle. Among these factors, the topography is flat to gently undulating, with slopes from 0% to 2%; the types of soil in this area are loam and clay, which have similar soil moisture contents [30,51]; and the geometric structures of the same crops in this area are similar. According to NOAA Global Surface Summary of Day [52], during the period of data acquisition, the average temperature in this area is 20.3 °C, and the accumulated precipitation is 4.5 cm. Therefore, the large variation of the incidence angle (21.2°–64.3°) is considered in this paper to be the main factor causing the large difference in backscattering between the near and far ranges of the same crop. In addition, the shape of cropland in this region is relatively regular, which makes it more conducive to study the impact of the incidence angle effect.

Previous works [20,53,54] have proven that among the polarization channels of HH, HV, and VV, the HV polarization is the most effective channel for discriminating different types of crops in the crop mapping of polarimetric SAR time-series images because the HV is remarkably sensitive to the volume scattering, which supports the value of structural features in distinguishing across crop types [20]. Therefore, only the backscattering coefficient of HV polarization ($S_{HV}$) was used for the crop mapping experiments in this paper; that is, the time-series curve for each sample $S_i$ was constructed using the corresponding $S_{HV}$ values in all 12 images (i.e., $S_i = \left\{ S_{HV}^{t1}, S_{HV}^{t2}, \cdots, S_{HV}^{tn} \right\}$). In addition, using only $S_{HV}$ rather than all the polarimetric features can simplify the analysis of the characteristics of time-series curves and provide guidance for crop mapping using single and dual-polarization data.

## 3. Methods

The constrained clustering methods using instance-level constraints were introduced to improve the reliability and stability of time-series clustering results. However, the effectiveness of pairwise constraints from the commonly used random sampling remains uncertain. Therefore, considering the characteristics of UAVSAR time-series images, a new method for generating informative pairwise constraints—namely, "active pairwise constraint learning (APCL)," is proposed. The flowchart of APCL is shown in Figure 4, which comprises the following two main steps: (1) extraction of the candidate sample set and (2) construction of the pairwise constraints.

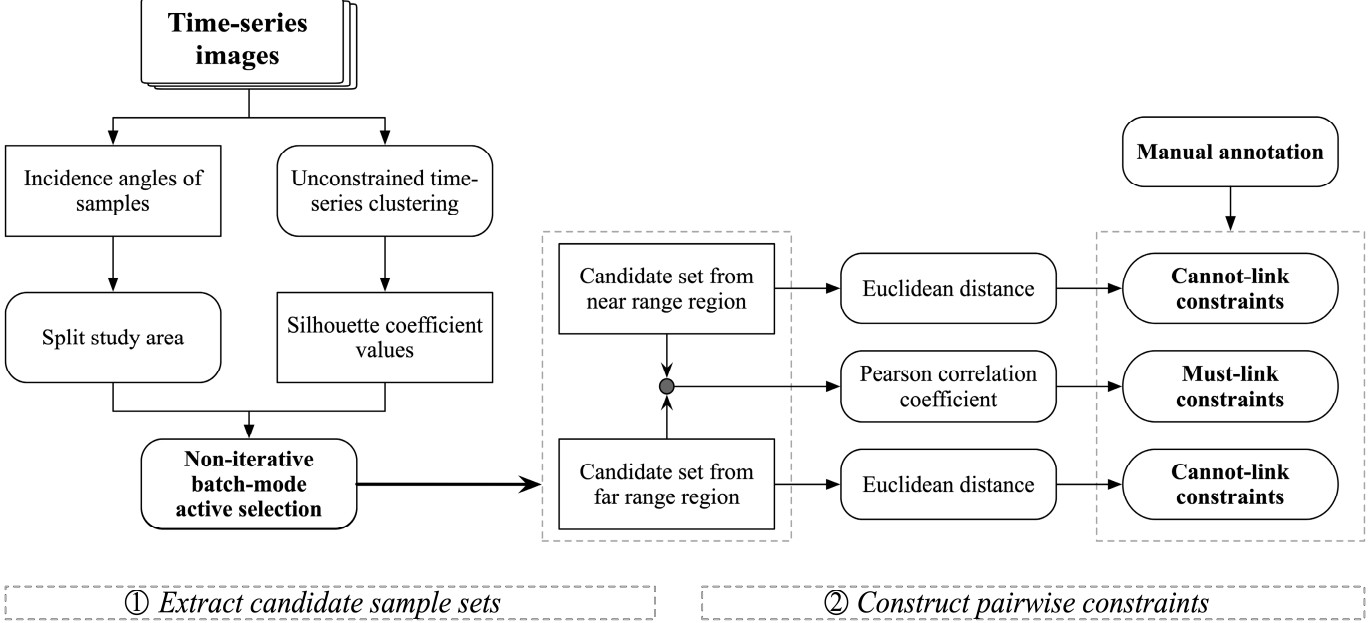

**Figure 4.** Flowchart of the proposed active pairwise constraint learning method.

### 3.1. Non-Iterative Batch-Mode Active Selection Theory

The construction of instance-level pairwise constraints is based on a group of samples with rich information (that are representative and diverse). Active learning is commonly used in classification tasks to select the informative samples, which are regarded as the most helpful to improve the classifier performance [55]. Two criteria are usually available for the measurement of the information of samples: (1) an uncertainty criterion and (2) a diversity criterion. The uncertainty criterion usually selects the samples near the class boundary because they are not easily categorized into a certain class by the classifier—that is, these samples have the highest uncertainties. Significant differences between a group of samples in the diversity criterion can effectively avoid knowledge redundancy; therefore, these samples have abundant information. The uncertainty and diversity criteria are considered in the proposed method to extract an informative candidate sample set.

However, conventional active learning techniques select only a single sample at each iteration for manual labeling and retrain the prediction model [56–59], which is inefficient and inapplicable to the clustering task of the paper. The batch-mode active learning (BMAL) technique is considered for selecting samples, in which a batch of samples are selected for manual annotation simultaneously at each round, to address the aforementioned limitations. The key issue of BMAL is to select the most informative batch of samples with as minimal redundancy to provide the most possible information to the prediction model [56,58,60,61]. Notably, differently from the conventional BMAL in classification tasks, a non-iterative batch-mode active selection scheme, in which the sample selection process was performed only once, is considered.

### 3.2. Extraction of the Candidate Sample Set

This step aims to extract a group of samples with representativeness and diversity based on the non-iterative batch-mode active selection scheme.

First, based on the study of Stanitsas et al. [62], the silhouette coefficient [63] is used for the measurement of the uncertainty of samples. The silhouette coefficient of a sample is calculated as follows:

$$S(i) = \frac{b(i) - a(i)}{max(a(i),\ b(i))} \tag{1}$$

where $a(i)$ is the average dissimilarity of sample *I* with all the other samples within the same cluster, and $b(i)$ is the lowest average dissimilarity of sample $i$ to each of the clusters to which it is not assigned. The range of $S(i)$ is between $-1$ and 1.

The samples with low silhouette coefficients are likely to be assigned to the wrong clusters; therefore, these samples will also have the highest uncertainties. The samples were first clustered using unconstrained time-series clustering, and then their silhouette coefficients were calculated on the basis of the clustering results to calculate the silhouette coefficient of each sample.

Two regions with equal space spans were then separated from the study area along the range direction: (1) the near range region (with incidence angles ranging from 21.2° to 47.2°, denoted by NR) and (2) the far range region (with incidence angles ranging from 54.0° to 64.3°, denoted by FR). The variation in the incidence angle values along the range direction is nonlinear; therefore, the corresponding incidence angle ranges of the two regions are different. The separation of NR and FR aims to extract two candidate sample sets (denoted by $U_{NR}$ and $U_{FR}$, respectively) independently from the two regions and then improves the informativeness of the generated constraints, which is comprehensively discussed in Section 4.4.

Finally, based on the silhouette coefficients of the samples, two candidate sample sets with high uncertainty and considerable diversity were extracted from the two regions using a greedy strategy. The main ideas of the strategy are as follows: (1) the samples with low silhouette coefficients are preferred, (2) the selected samples should be evenly distributed in different clusters, and (3) the geometric distance in the feature space between samples inner a cluster should be larger than a specified threshold T. The first idea is to ensure that the selected samples have high uncertainty, and the last two ideas aim to ensure that the selected samples are of considerable diversity. In real programming implementation, achieving the greedy sample extraction strategy is easy, and various implementation methods are available [37]; however, its implementation is not comprehensively discussed. In the proposed method, the sizes of $U_{NR}$ and $U_{FR}$ are both set to one-third of the number of constraints to be constructed.

K-means [64] is used as the unconstrained clustering algorithm, with the ED as the similarity criterion, to visualize the efficiency of the proposed method. The extracted candidate sample sets $U_{NR}$ and $U_{FR}$ are shown in Figure 5, where the number of samples in each candidate sample set is 150.

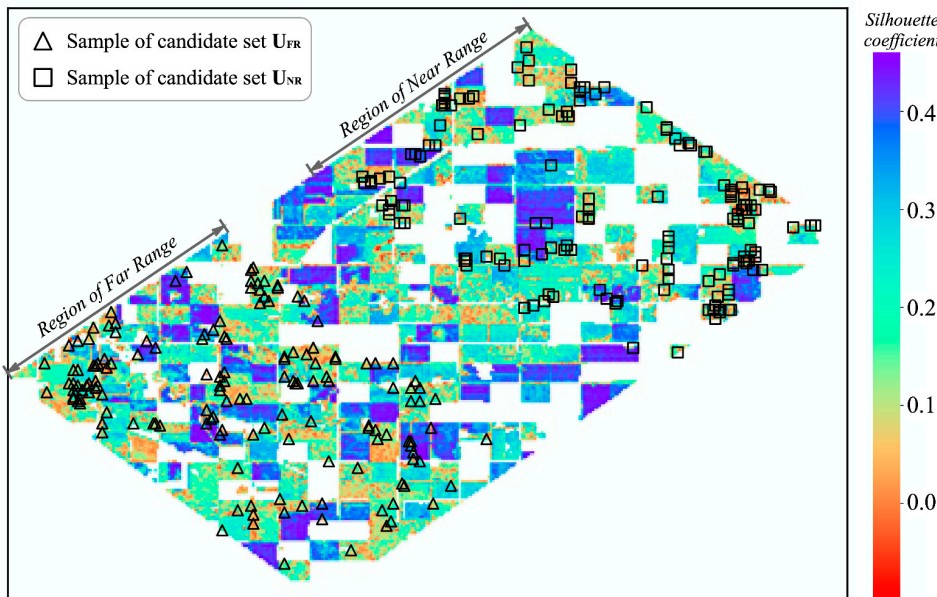

**Figure 5.** Visualization of the distribution of the candidate sample sets. The background image is the heatmap of the samples' silhouette coefficients.

### 3.3. Construction of the Pairwise Constraints

Two characteristics of the crops in the UAVSAR time-series images are considered when learning the pairwise constraints from $U_{NR}$ and $U_{FR}$: (1) the time-series curves of the same type of crops from the near and far ranges may have high shape similarity but large spatial distances; and (2) the time-series curves of the different types of crops with small differences of incidence angle may be close to each other in the feature space, thereby leading to misclassification. Consequently, the MLCs and CLCs are constructed on the basis of the first and second characteristics, respectively. The detailed procedure of constructing the $N_{MLC}$ MLCs and $N_{CLC}$ CLCs is presented below.

#### 3.3.1. Construction of the Must-Link Constraints (MLCs)

First, the Pearson correlation coefficient is used to construct a shape similarity matrix $M_{Pearson}$:

$$M_{Pearson}(i,j) = r\left(s^i_{NR}, s^j_{FR}\right) \tag{2}$$

where $s^i_{NR}$ and $s^j_{FR}$ are the $i$th sample of $U_{NR}$ and the $j$th sample of $U_{FR}$, respectively. Thus, each element in $M_{Pearson}$ corresponds to two samples from $U_{NR}$ and $U_{FR}$.

Second, the elements in $M_{Pearson}$ are sorted in accordance with their Pearson correlation coefficient values from small to large. Then, the pairs of samples, wherein the samples in each pair are of the same crop type according to the order are chosen until $N_{MLC}$ pairs of samples, are selected. These selected sample pairs are the required MLCs. Furthermore, each sample can be selected at most two times to guarantee the diversity of the constraints.

The learned MLCs have the following features. (1) Each pair of constraints links the samples of the same class from different incidence angle regions (which means that their backscattering coefficients markedly differ). (2) The shape similarity between the time-series curves of linked samples is low (because their correlation coefficients are small). (3) The linked samples are from the candidate sample set (indicating their high clustering uncertainties). Therefore, the linked samples are the most difficult to cluster correctly.

#### 3.3.2. Construction of the Cannot-Link Constraints (CLCs)

Two sets of CLCs are constructed from $U_{NR}$ and $U_{FR}$. Taking the construction of the CLCs from $U_{FR}$ as an example, the detailed process is as follows.

Firstly, the samples in candidate set $U_{FR}$ are used to construct an ED matrix $M_{ED}$:

$$M_{ED}(i,j) = ED\left(s^i_{FR}, s^j_{FR}\right) \tag{3}$$

where $s^i_{FR}$ is the $i$th sample of $U_{FR}$, $s^j_{FR}$ is the $j$th sample of $U_{FR}$, and $M_{ED}(i,j) = M_{ED}(j,i)$. That is, $M_{ED}$ is a symmetric matrix.

The elements in the upper triangular matrix are then sorted in accordance with their ED values from small to large, where each element corresponds to a pair of samples from $U_{FR}$.

Finally, the $N_{CLC}/2$ pairs of samples, which are different types of crops, were selected in accordance with the order and then used as a set of CLCs. In addition, each sample was selected at most two times to guarantee the diversity of the constraints. Another set of CLCs of quantity $N_{CLC}/2$ was constructed from $U_{NR}$ using the above procedure, and a group of CLCs of quantity $N_{CLC}$ can then be obtained.

The linked samples in the above flowchart are from the same region (region FR or NR) and are close in the feature space, allowing easy clustering into the same group. Therefore, using these samples as CLCs could help improve the clustering.

K-means was used as the unconstrained clustering algorithm and the ED as the similarity measure for the active constraint learning to visualize the distribution of the constraints constructed by the proposed method (APCL). The constraints of another two constraint learning methods were used for comparison. The first method was random sampling based on regions of incidence angle (denoted by RSRIA), which randomly generates CLCs

within the same incidence angle region and generates MLCs between different incidence angle regions. The second method is the commonly used random sampling. The number of each kind of pairwise constraint was set to 200 for both constraint learning methods. The distributions of the obtained constraints are shown in Figure 6. Compared with the constraints of random sampling, the constraints of the proposed method link the samples with low silhouette coefficients, and all these samples are from either the near or the far range.

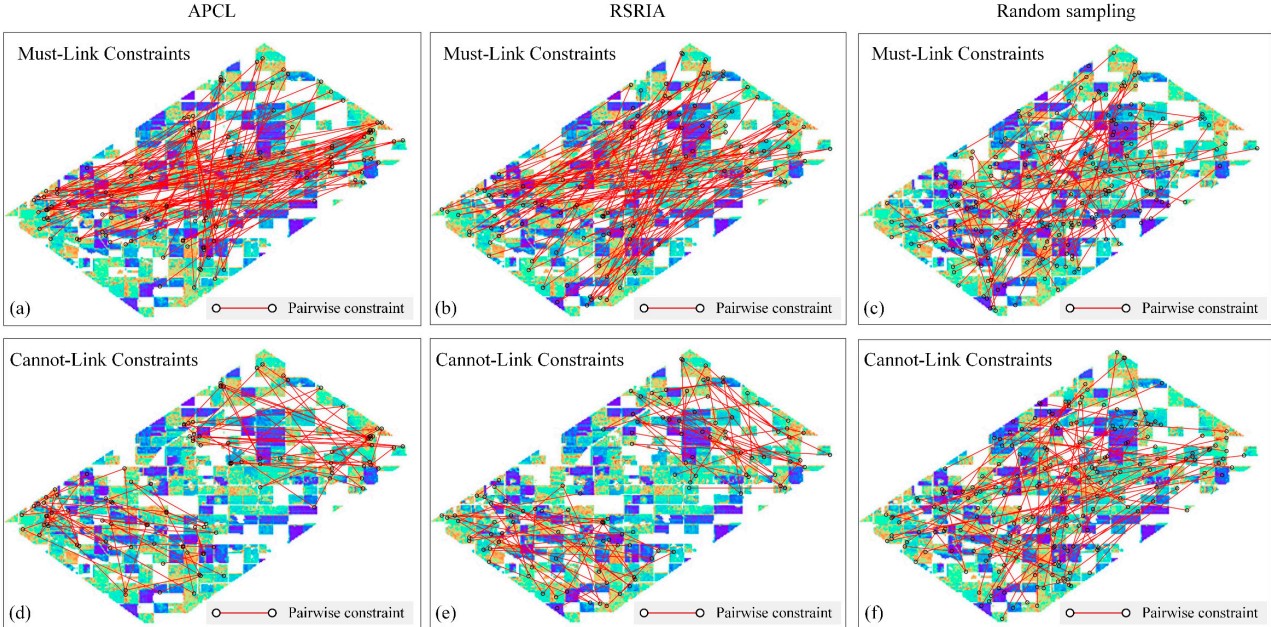

**Figure 6.** Distribution of the pairwise constraints, where the background is the heatmap of the samples' silhouette coefficients: (**a**) MLCs of APCL; (**b**) MLCs of RSRIA; (**c**) MLCs of random sampling; (**d**) CLCs of APCL; (**e**) CLCs of RSRIA; (**f**) CLCs of random sampling.

## 4. Results and Discussion

This section includes four parts. In the first part, a qualitative analysis of impact of the incidence angle effect on crop separability is presented via the time-series curves of various crop types with different incidence angles. In the second part, we present the results of numerous experiments that were conducted to illustrate the efficiency of the proposed method. Only 20% of the total number (30,605) of samples of the time-series images were used in each experiment due to the large number of experiments and the time-consuming time-series clustering. In the third part, all the samples in the study area were used for the clustering, and the corresponding crop classification maps were then generated. In the experiment, when generating MLCs and CLCs, the processes of identifying whether the two selected samples should be linked were based on the ground truth data (in practice, this method should be conducted manually). In the fourth part, a quantitative evaluation of the informativeness of the constraints was conducted. All the experiments were performed on a 64-bit Window 10 PC, and the programming language was Python 3.

### 4.1. Impact of the Incidence Angle Effect on Crop Separability

First, considering that the incidence angle has different impacts on different regions, the following three regions with equal space spans were separated from the study area along the range direction to further discuss the temporal characteristics of the crop under different incidence angles: a region with small incidence angles (incidence angles ranging from 21.2° to 32.0°, denoted by Inc_S); a region with medium incidence angles (incidence angles ranging from 50.0° to 56.0°, denoted by Inc_M); and a region with large incidence

angles (incidence angles ranging from 60.0° to 64.3°, denoted by Inc_L). One hundred samples of each type of crop were then sampled from these three regions, as follows:

$$S_{HV} = \frac{1}{100} \sum_{i=1}^{100} S_{HV}^i \tag{4}$$

The averaged $S_{HV}$ of the selected samples of each crop was used to construct the time-series curves. A total of 12 time-series curves were generated for the four crops from the three regions, as shown in Figure 7.

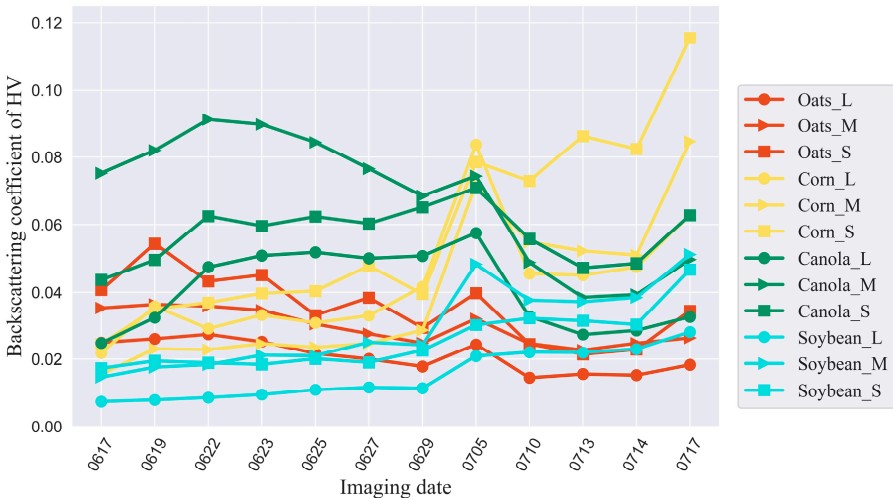

**Figure 7.** Time-series curves of the crops in different regions. "_L" denotes that the crop is from the region of Inc_L, "_M" denotes that the crop is from the region of Inc_M, and "_S" denotes that the crop is from the region of Inc_S.

These findings show that the $S_{HV}$ values of all the crops substantially change over time. For oats, the shape of the time-series curve in the region of Inc_L is similar to that in the region of Inc_M, and that in the region of Inc_S markedly changes. For corn, the $S_{HV}$ of the time-series curve in Inc_S is significantly higher than those of Inc_M and Inc_L after day 0705. For canola, the $S_{HV}$ of the time-series curve in Inc_M is significantly higher than those of Inc_S and Inc_L before day 0629. For soybean, the shapes of the three time-series curves are similar. Thus, the different characteristics of the time-series curves of some crops are attributed to the different incidence angles.

Three similarity matrices based on the Euclidean distance (ED), dynamic time warping (DTW), and Pearson correlation coefficient were generated as shown in Figure 8 to further evaluate the distinguishability of these crop time-series curves. The ED is a lock-step measure (one-to-one) and sensitive to scaling, and the Pearson correlation coefficient is invariant to the scale and location of the data points [65]. DTW [66] uses a dynamic programming (DP) method to find an optimal alignment between the two time series via nonlinear warping, thus increasing its suitability for evaluating the similarity of time series of different lengths.

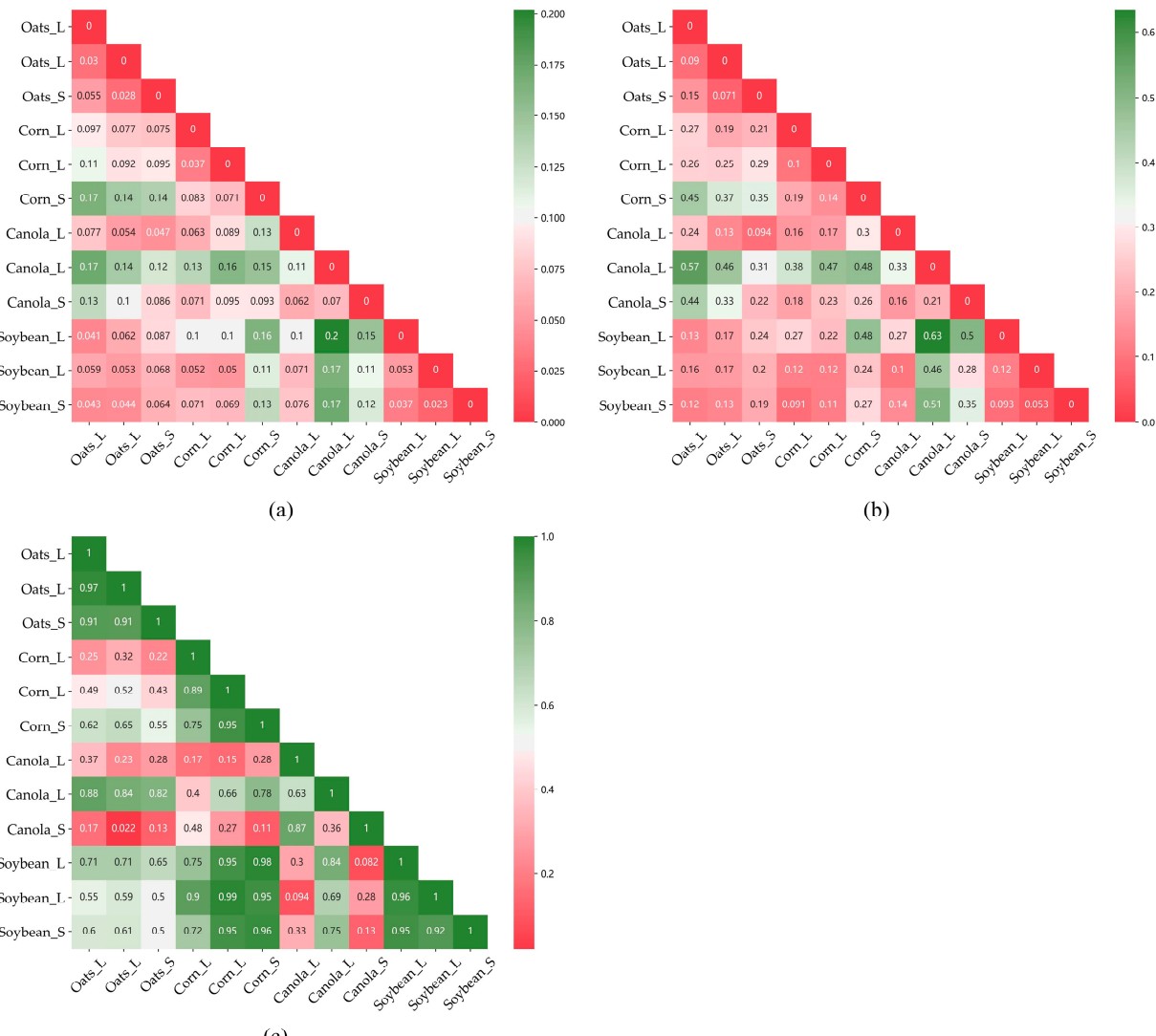

**Figure 8.** Similarity matrices for the crop time-series curves: (**a**) similarity matrix based on the ED, where 0 denotes the highest similarity; (**b**) similarity matrix based on the DTW, where 0 denotes the highest similarity; (**c**) similarity matrix based on the Pearson correlation coefficient, where 1 denotes the highest similarity.

These findings show that the similarities between some curves of different crops are relatively high. Therefore, many crop samples will be easily misclassified in the time-series clustering. For instance, in Figure 8a, the ED between Oats_L and Oats_S is larger than that between Oats_L and Soybean_L and Soybean_S. Thus, when using the ED as the similarity metric in time-series clustering, the samples of oats in Inc_L are likely to be clustered into one group with the soybean samples. The ED between the soybean samples and their time-series curves is relatively small, indicating that they are less likely to be misclassified in the process of time-series clustering. Figure 8c shows that the similarities between corn and soybean are relatively high, indicating that the samples of these crops would be easily misclassified when using Pearson correlation coefficient as the similarity metric.

The characteristics of the crops in the UAVSAR time-series images can be summarized as follows. (1) The time-series curves of crops change significantly with time. (2) The time-series curves of the same type of crops have similar shapes in all the regions of Inc_S, Inc_M, and Inc_L. (3) The spatial distance between the time-series curves of the same crops in the feature space may be substantially large due to their difference in incidence angle. (4) The time-series curves of different crops may have considerable shape similarities

or small spatial distances. Among these characteristics, the first two characteristics are beneficial to the clustering of the same crops from different regions, and the latter two characteristics are not conducive to the clustering.

### 4.2. Constrained Clustering Accuracy Analysis

The experimental settings of each group of experiments in the first part are presented in Table 3. Each group of experiments was repeated five times, and the average accuracy was taken as the final result for analysis. In each group of experiments, 20% of the total samples were first randomly sampled, and then these samples were used in the experiments to make the results comparable. Finding the optimal number of initial clustering centroids is difficult for all scenarios because it may be different under various clustering methods, similarity criteria, and types of constraints. The optimal number was simply set as eight (two times the number of crop types), and the number of MLCs was set similarly to that of the CLCs.

**Table 3.** Experimental settings of each group of experiments.

| | |
|---|---|
| Constrained Clustering Algorithm | 1. COP-KMeans, 2. PC-KMeans, 3. MIP-KMeans |
| Time-series similarity criterion | 1. Euclidean distance (ED) 2. Dynamic time warping (DTW) 3. Pearson correlation coefficient (Pearson) |
| Constraint generation method | 1. Active pairwise constraint learning (APCL) 2. Random sampling based on regions of incidence angle (RSRIA) 3. Random sampling |
| Number of pairwise constraints | {0,150,300,450,600,750,900,1050,1200,1350,1500,1650,1800,1950,2100,2250,2400} |
| Values of sample's time series | Backscattering coefficients of HV polarization in the 12 images |

The three constrained clustering algorithms used in the experiments are all K-means-based methods, which are suitable for time-series application because they iteratively calculate distances to cluster and update these centroids [42]. Their iterative processes are similar, but their usage of constraints is quite different. Therefore, the use of these methods is helpful to analyze how the constraints work and evaluate the effectiveness of constraints from different methods.

In COP-KMeans [39], before assigning a sample to a cluster, it first judges whether the sample is linked with clustered samples by the MLCs (if any) and assigns the sample to the same cluster; otherwise, it is assigned to the nearest cluster while not violating the CLC. However, if not all the constraints are satisfied, then the algorithm fails and no clustering result can be obtained. The following adjustments are performed in the experiments to ensure the COP-KMeans output clustering result: if a sample has constraint violations in each iteration, then the sample will not be assigned to any clusters; if samples that have constraint violations still exist at the end of the last iteration, then these samples will be assigned to the clusters with the fewest constraint violations. PC-KMeans [67] formulates the goal of clustering as minimizing a combined objective function: the sum of the total distance between the samples and their cluster centroids and the cost of violating the pairwise constraints. The constraint violation cost was a relative value in the experiments; that is, the cost of each constraint violation was set as 0.1 times the distance between the sample and the cluster centroid. MIP-KMeans [68] considers the sample constraints and objective function in its entire combinatorial configuration and uses a general optimization tool (the Gurobi optimizer) to solve it [42]. The maximum number of iterations of these

algorithms was set as 200, and the convergence condition for these methods was that the change of position of any centroids is no more than a threshold (which was set as 0.0005).

Three time-series similarity measures were used in this study: the ED, DTW, and the Pearson correlation coefficient (denoted by Pearson). For each experiment, in the unconstrained clustering for calculating silhouette coefficient, the used time-series similarity measure was set to be the same as that used in the constrained clustering.

The normalized mutual information (NMI) [69] was used as the clustering evaluation index. The range of NMI is [0, 1], and a high value indicates an effective clustering performance. The mean and standard deviation of NMI for each group of experiments are shown in Figure 9.

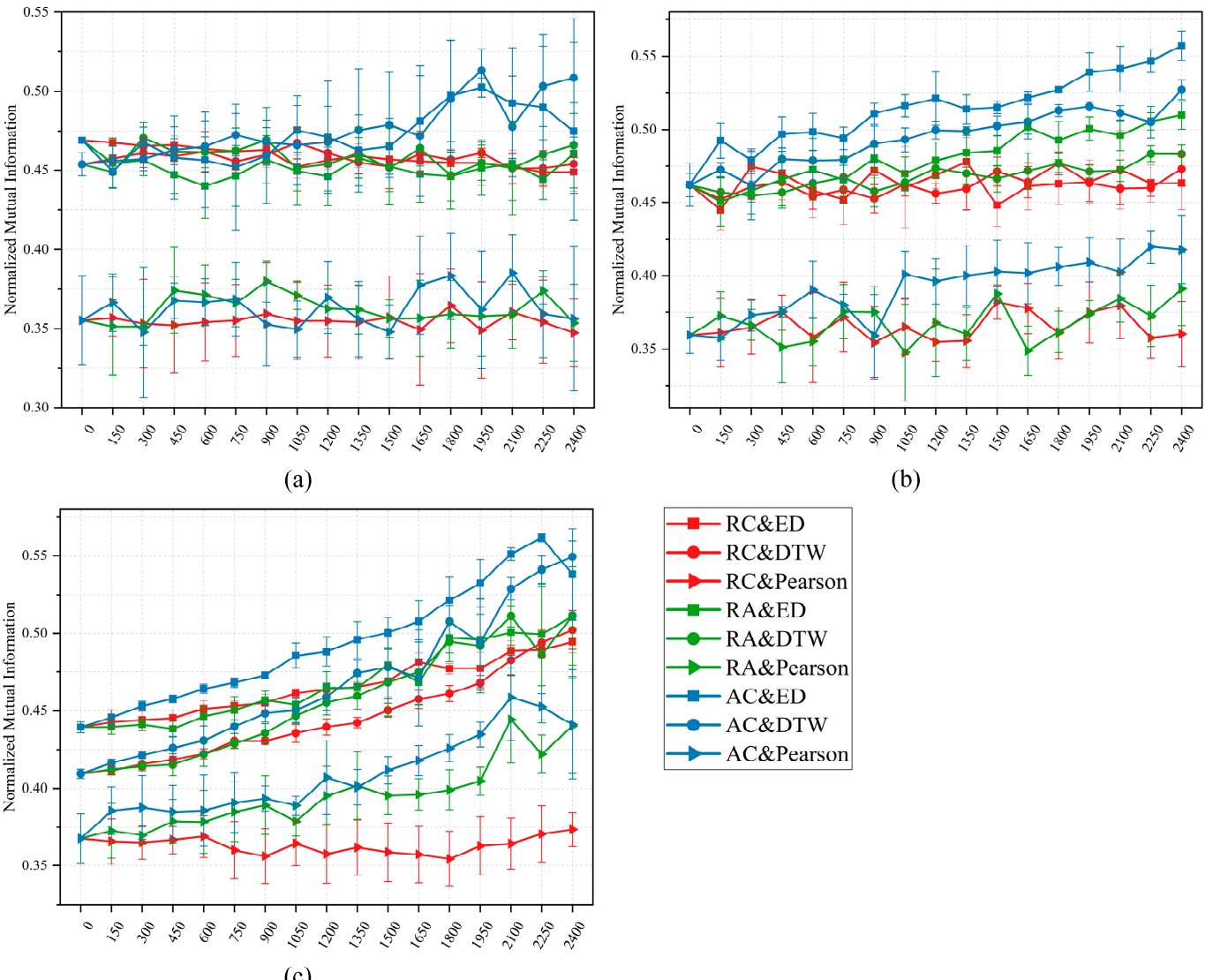

**Figure 9.** Performance plots show how the NMI increases with the number of pairwise constraints: (**a**) results of COP-KMeans; (**b**) results of PC-KMeans; (**c**) results of MIP-KMeans. The circular markers, the inverted triangular markers, and the square markers correspond to the results obtained using ED, DTW, and Pearson, respectively.

For convenience, in Figure 9, "RC" represents "constraints from random sampling," "RA" represents "constraints from RSRIA," and "AC" represents "constraints from APCL." For example, "RC&ED" represents the results obtained using constraints generated by random sampling and the ED as the similarity criterion.

As shown in Figure 9, the results are for unconstrained clustering when the number of the constraints is zero. The clustering processes of the three constrained clustering algorithms are consistent with that of the traditional K-means method when the number is zero and their NMI values are different. This phenomenon is due to the random selection of the initial clustering centroids of COP-KMeans and the generation of initial clustering centroids of PC-KMeans and MIP-KMeans in specific ways. For further details, please refer to [67,68].

These findings show that COP-KMeans is insensitive to the number of constraints, and only "AC&ED" and "AC&DTW" have better NMI than unconstrained clustering results when the number of pairwise constraints is larger than 1650. PC-KMeans can effectively utilize the constraint information; therefore, its accuracies are generally better than those of the two other algorithms. Among the results of PC-KMeans, the NMI curves of APCL are better than random sampling and RSRIA and all have an ascending trend, especially for the curve of "AC&ED." MIP-KMeans is remarkably sensitive to the number of constraints. Thus, all of the NMI curves show an ascending trend with the increase in the number of pairwise constraints, except for the curve of "RC&Pearson." In addition, when the number of constraints was 2250, the curve of "AC&ED" of MIP-KMeans achieved the best NMI value of 0.562 among all the results in Figure 9.

Overall, the experimental results demonstrate the superiority of the proposed APCL, which can improve the clustering accuracy effectively under various constrained clustering algorithms, time-series similarity criteria, and different numbers of constraints.

No obvious correlation exists between the running time of the clustering algorithms and the number of constraints in the above experiments. Therefore, the average running time of each algorithm was calculated under all numbers of constraints, as shown in Figure 10.

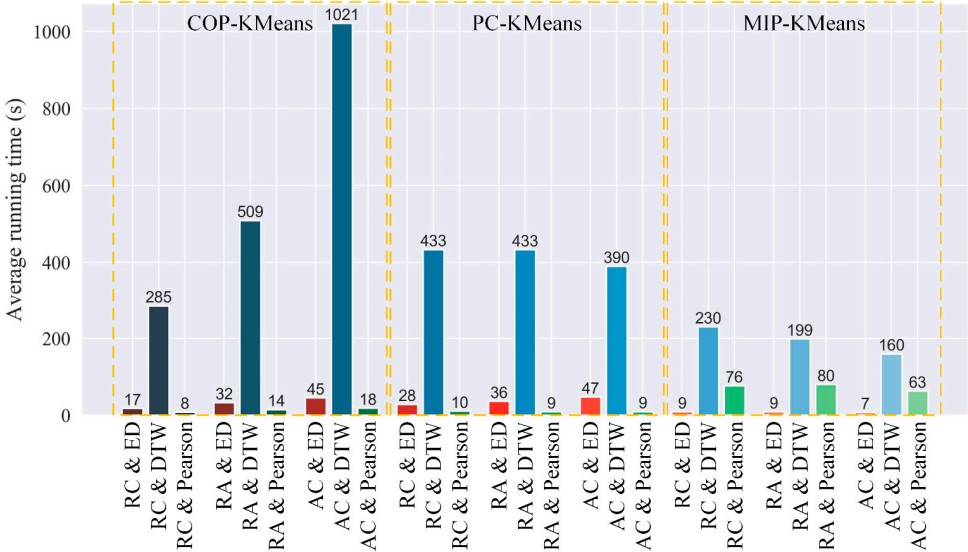

**Figure 10.** Average running time of each algorithm. The bars in the red range are the results using ED, the bars in the blue range are the results using DTW, and the bars in the green range are the results using Pearson.

The running times of different algorithms and time-series criteria are quite different. DTW is more time-consuming than ED and Pearson among all the methods due its high computational complexity. COP-KMeans with AC&DTW is the most time-consuming method because it requires additional iterative time to convergence. When using ED or DTW as the similarity measure, MIP-KMeans uses less running time than COP-KMeans and PC-KMeans. This finding is due to MIP-KMeans solving an overall optimal result in each iteration [68], which demonstrated a fast convergence speed, resulting in fewer

iterations than the other two algorithms. However, in the results of using Pearson as the similarity measure, the running time of MIP-KMeans is significantly larger than COP those of KMeans and PC-KMeans. The reason is that the distinguishability of various crops was poor when using Pearson. Thus, MIP-KMeans had to spend additional time in performing global optimization in each iteration to find an overall solution in which the pairwise constraints were also satisfied. Therefore, the convergence speed of MIP-KMeans is slow, resulting in higher time consumption than the two other algorithms.

### 4.3. Crop Classification Accuracy Analysis

All the samples (the total number is 30,605) of the study area were used for time-series clustering, and PC-KMeans was used as the constrained clustering algorithm to further evaluate the effectiveness of APCL. The three time-series similarity criteria (ED, DTW, and Pearson) and the generation methods (RSRIA and random sampling) of the two constraints were also used for comparison. The total number of MLCs and CLCs was 12,000 (40% of the total samples), and the number of initial cluster centroids was set to eight. In addition, the results of PC-KMeans with zero constraints were used as the baseline for comparison.

After obtaining the clustering results, for each cluster, the number of samples of each class was counted in accordance with the reference map, and then the class label with the largest number of samples was assigned to the cluster (in practice, the process of labeling clusters should be performed manually). Each experiment was repeated five times. The accuracy of each result is presented in Table 4. The classification map with the highest accuracy for each method is presented in Figure 11.

**Table 4.** Classification accuracy obtained by all methods.

| Method | Similarity Criteria | F-Score | | | | OA | Kappa |
| --- | --- | --- | --- | --- | --- | --- | --- |
| | | Oats | Corn | Canola | Soybean | | |
| Unconstrained | ED | 71.8% | 76.5% | 86.1% | 88.2% | 83.5% | 76.6% |
| | DTW | 66.9% | 73.2% | 82.6% | 88.0% | 81.8% | 73.3% |
| | Pearson | 68.4% | 52.5% | 83.3% | 75.9% | 73.5% | 62.4% |
| Random sampling | ED | 69.5% | 78.2% | 84.3% | 89.7% | 83.6% | 76.6% |
| | DTW | 66.4% | 75.9% | 82.9% | 88.3% | 82.3% | 74.1% |
| | Pearson | 67.1% | 52.9% | 84.4% | 81.9% | 77.2% | 66.6% |
| RSRIA | ED | 74.4% | 80.4% | 87.6% | 91.1% | 86.2% | 80.3% |
| | DTW | 71.3% | 79.3% | 85.7% | 89.4% | 84.7% | 77.6% |
| | Pearson | 66.4% | 36.9% | 84.0% | 82.0% | 75.8% | 63.9% |
| APCL | ED | 75.9% | **85.5%** | 89.2% | **92.0%** | **88.1%** | **82.9%** |
| | DTW | 81.9% | 83.6% | **90.0%** | 89.9% | **88.1%** | 82.8% |
| | Pearson | **82.5%** | 51.0% | 87.1% | 82.2% | 79.8% | 70.2% |

The four methods unconstrained clustering, random sampling, RSRIA, and APCL all achieved their best accuracy (kappa coefficient) when using ED as the similarity criterion, which were 76.6%, 76.6%, 80.3%, and 82.9%, respectively, as shown in Figure 11. APCL also achieved a kappa of 82.8% when using DTW. All the four methods had poor accuracies when using Pearson. However, the accuracy of APCL (kappa = 70.2%) was still higher than those of the other methods.

Among the four crop types, the F-score for oats was relatively low when using ED as the similarity criterion because its backscattering coefficients were close to that of other crops, as shown in Figure 7. When using Pearson as similarity criterion, the F-score for corn was substantially low for all the methods. This phenomenon was due to the relatively low shape similarity of corn's time-series curves in different incidence angles, increasing the difficulty of clustering into one group using Pearson. In addition, for the F-score of each crop type, the best accuracies of oats, corn, canola, and soybean were all achieved by APCL.

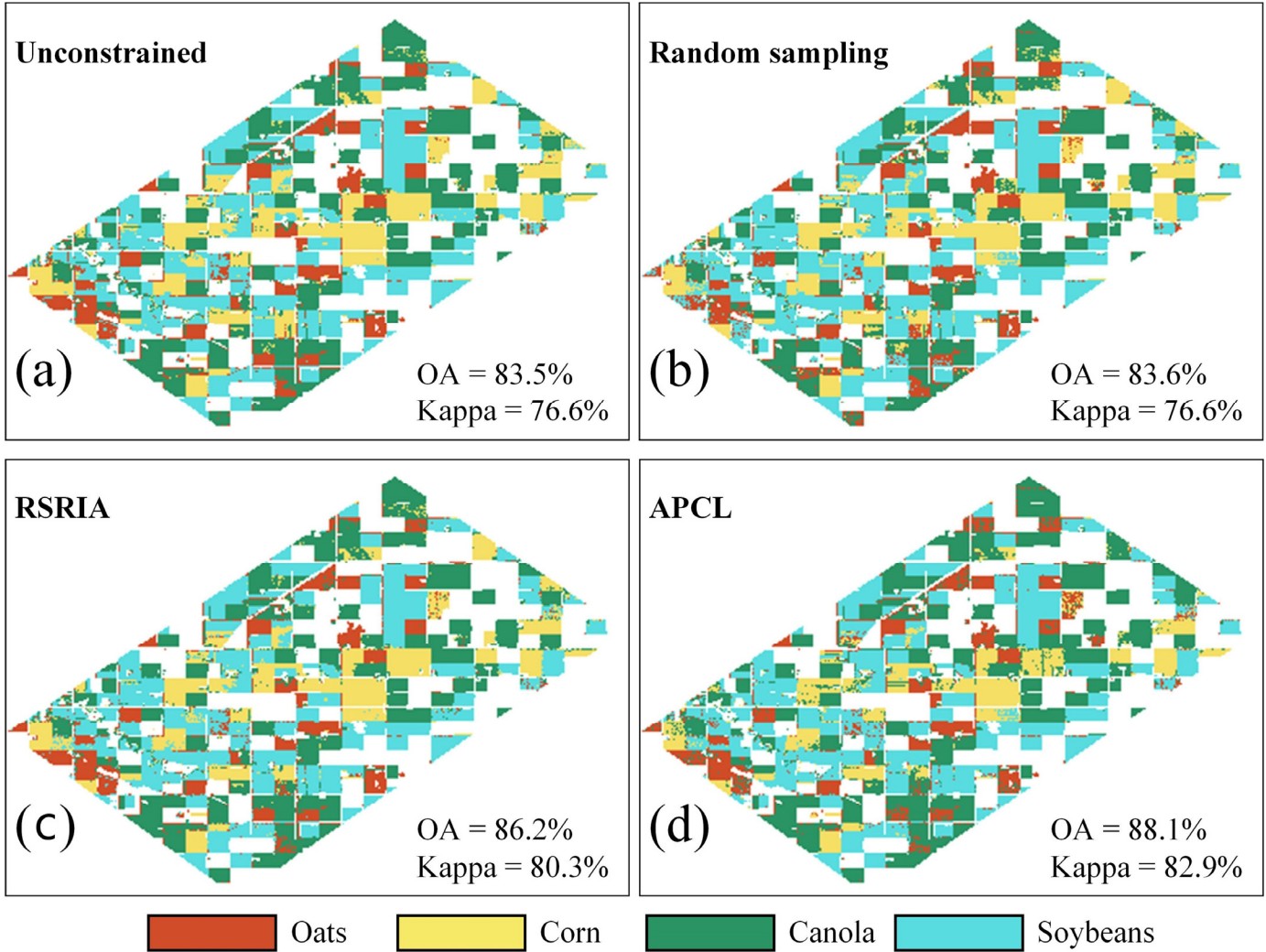

**Figure 11.** Crop classification maps with the highest accuracy for each method: (**a**) unconstrained clustering; (**b**) random sampling; (**c**) RSRIA; (**d**) APCL.

When the results of unconstrained clustering were used as the baseline, the constraints of random sampling minimally helped improve the accuracy, and those of RSRIA can improve the accuracy to a certain extent. This phenomenon demonstrates the effectiveness of the proposed strategy of generating CLCs within the same incidence angle region and generating MLCs between different incidence angle regions. The comparison between the results of RSRIA and APCL shows the significance of the proposed strategy of extracting candidate sample set based on silhouette coefficients. These results prove the performance and application value of APCL in the constrained clustering of Airborne SAR imagery.

### 4.4. Evaluation of the Informativeness of the Constraints

If the constraints can be satisfied in the unconstrained clustering results, the constrained clustering based on these constraints may not improve the clustering accuracy [36]. In other words, for a group of constraints, a large number of unsatisfied constraints in the unconstrained clustering result indicate that these constraints contain additional information. Therefore, based on the results of unconstrained clustering with PC-KMeans as the clustering method and ED as the time-series similarity criterion, the quantity of information for the pairwise constraints (6000 pairs of MLCs and 6000 pairs of CLCs) obtained by random sampling, RSRIA, and APCL was evaluated. The number of unsatisfied constraints in the unconstrained clustering results is presented in Table 5.

**Table 5.** Number of unsatisfied constraints in the unconstrained clustering results.

| Method | Must-Link Constraint | Cannot-Link Constraint | Total |
|---|---|---|---|
| Random sampling | 3089 | 308 | 3397 |
| RSRIA | 3302 | 304 | 3606 |
| APCL | 4283 | 1691 | 5974 |

First, the total number of unsatisfied constraints for APCL is larger than that for the other methods. Second, only a few of the CLCs of random sampling and RSRIA were unsatisfied, and many of the CLCs of APCL were unsatisfied. Therefore, the information contained in the constraints of APCL is abundant. In addition, the number of unsatisfied MLCs is substantially larger than that of the unsatisfied CLCs, which indicates that the MLCs carry additional information and are thus effective at improving the accuracy of constrained clustering. Thus, before the constraint learning, the informativeness of the different kinds of constraints should first be evaluated, and the specific constraints should then be constructed to acquire substantially useful information under limited size of constraints [38,42].

In the following, the reason for having additional unsatisfied MLCs in APCL is discussed. The clustering is markedly affected by the different incidence angles. Thus, the absolute value of the incidence angle difference between every two samples in an MLC is calculated, and then the histograms are drawn, as shown in Figure 12. Each bar in Figure 12 corresponds to the number of satisfied or unsatisfied MLCs under the specific difference of incidence angle.

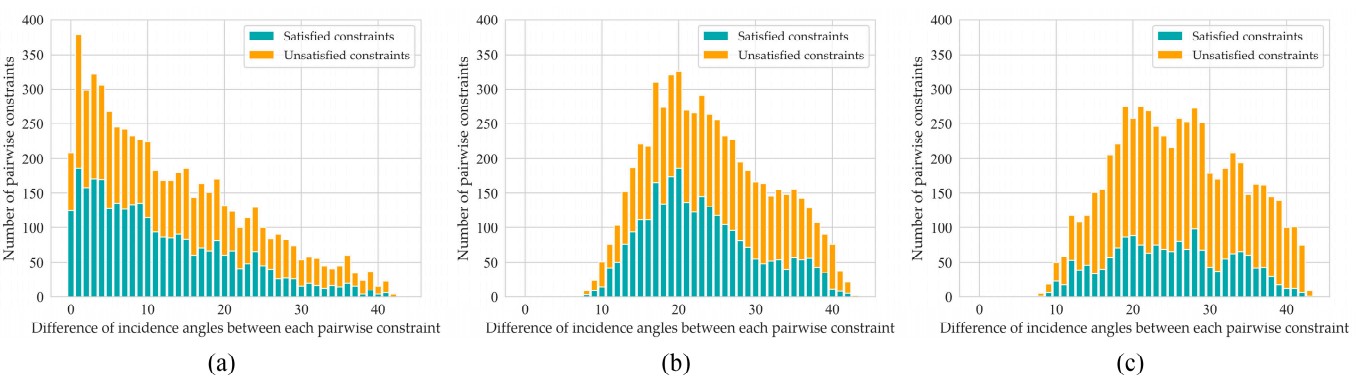

**Figure 12.** Numbers of satisfied and unsatisfied must-link constraints under the absolute values of the incidence angle differences: (**a**) random sampling; (**b**) RSRIA; (**c**) APCL.

In Figure 12a, most of the MLCs of random sampling show a small incidence angle difference, and half are satisfied. In Figure 12b, most of the MLCs have in an incidence angle range of 13–35°, but many remain unsatisfied. In Figure 12c, most of the MLCs have an incidence angle range of 15–39°, and the number of satisfied MLCs is significantly less than in the above results. Thus, MLCs are only minimally satisfied when the difference between incidence angles is large. This phenomenon is due to the differences in backscattering coefficients of the same crop's samples with large differences in incidence angles. Therefore, clustering these samples into one group by unconstrained clustering is difficult. Second, the proposed APCL can generate MLCs that are difficult to satisfy. The above results prove that the pairwise constraints obtained by the proposed method are more informative than that of RSRIA and random sampling.

As a semi-supervised clustering technique, constrained clustering also needs some supervised information to generate constraints. However, unlike the conventional supervised methods, supervision by instance-level constraints is more general and more realistic than specific class labels. By using knowledge, a user can specify whether pairs of samples

belong to the same cluster or not even when the class labels remain unknown [36]. The poor visual effect of PolSAR images increases the difficulty in identifying the class labels of samples [29,70]. In this case, the method of generating pairwise constraints shows its advantages. However, designing the corresponding human–computer interaction system in practical application to construct constraints remains a problem that must be solved in the future.

## 5. Conclusions

An active pairwise constraint learning method is proposed in this paper to reduce the incidence angle effect in crop mapping. The proposed method can construct informative pairwise constraints based on the incidence angles of the samples and a non-iterative batch-mode active selection scheme. Comparison experiments on the UAVSAR images have shown the effectiveness of the proposed APCL method compared to the commonly used constraint construction methods for constrained clustering at improving the crop mapping accuracy and the superiority. However, the proposed method ignores transferring pairwise constraint information between samples to reinforce the constraints, and the future work will focus on effectively transferring constraint information. Thus, a way to pre-evaluate the optimal proportion of different types of constraints is also needed.

**Author Contributions:** Conceptualization: X.Q. and J.Y.; methodology: X.Q., K.S. and L.Z.; validation: X.Q. and L.Z.; investigation: X.Q., B.W. and L.Z.; resources: J.Y. and P.L.; writing of original draft: X.Q.; writing—review and editing: L.Z., J.Y., X.Q., Y.X. and K.S.; supervision: P.L. and J.Y.; project administration: J.Y. and P.L.; funding acquisition: J.Y., P.L. and B.W. All authors have read and agreed to the published version of the manuscript.

**Funding:** This research was funded by the National Natural Science Foundation of China (Grant No. 61971318), the Strategic Priority Research Program of the Chinese Academy of Sciences (Grant No. XDA19030201), the Shenzhen Fundamental Research Program (Grant No. JCYJ20200109150833977), the Joint Funds of the National Natural Science Foundation of China (Grant No. U2033216), and the National Natural Science Foundation of China Major Program (Grant No. 42192580).

**Data Availability Statement:** The data presented in this study are openly available in (UAVSAR_POLSAR) at (10.5067/7PEQV8SVR4DM).

**Acknowledgments:** The authors would like to present their acknowledgments to the JPL NASA and SMAPVEX 2012 teams for providing the UAVSAR images, and the NASA National Snow and Ice Data Center Distributed Active Archive Center (NSIDC DAAC) for making and providing the 2012 crop map of the Winnipeg area.

**Conflicts of Interest:** The authors declare no conflict of interest.

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
