# Peer review of "Active Pairwise Constraint Learning in Constrained Time-Series Clustering for Crop Mapping from Airborne SAR Imagery"

_remotesensing, doi:10.3390/rs14236073_

Round 1

Reviewer 1 Report

General comments

1. What are the advantages and disadvantages of this study

2. In Fig.4, the time curves of crops are changed under different incident angles. A comparison tests where the incident angle do not change is needed to confirm the explain nation. 

3. Line 246: Can you explain how to select the angle range of near and far range region. Because the angle range classification seems different from that in section 2.2. 

Specific comments

1. Line 173: Edit image, improve quality.

2. Line 447: Enlarged figure are suggested, given the overall classification maps are similar. 

Reviewer 2 Report

This paper proposes a new approach named active pairwise constraint learning method (APCL) to improve the accuracy of crop mapping using 12 multitemporal airborne SAR data strongly affected by varying incidence angles (21–64 degrees). I have the following comments which can be rated as minor comments:

1. In L25 (Abstract), the most common meaning of UAV is unmanned aerial vehicle rather than uninhabited aerial vehicle.

2. In Abstract, please, provide the main quantitative result of the research showing the effectiveness of the proposed method.

3. Was there any reason to divide the range of incidence angles into 21.2°-32.0°, 50°-56°, and 60°-64.3°? Why the authors did not divide the range into, for example, three equal intervals? Same for the incidence angles ranging from 21.2°-47.2° and from 54.0° to 64.3° in P8.

4. I believe the Figures 4 and 5 (and their corresponding paragraphs) from section 2.2. Impact of the Incidence Angle Effect on Crop Separability are, in fact, part of the Results section.

5. In Figure 5, the upper part of the matrices is exactly equal to the lower part, in relation to the main diagonal. I suggest excluding the upper part of the matrices.

6. I suggest joining the Section 4. Experiments and Analysis and the Section 5. Discussion and naming it as 4. Results and Discussion.

7. In the Reference section, the Remote Sensing journal uses abbreviated form for the journal´s name (for example, Remote Sens. rather than Remote Sensing).

Reviewer 3 Report

Dear Authors,

The paper proposes a fascinating method, the active pairwise constraint learning method (APCL), for constrained time-series clustering to enhance crop mapping accuracy. Overall, the work is quite interesting. The introduction is very informative. The methods and results are well-explained, yet the discussion section is very shallow and looks like an extended results section. Therefore, I suggest reworking it. The conclusion section is well presented yet very broad, which can be easily improved by giving a summary of your findings, limitations of the work, and a future perspective on what is next. I also recommend injecting more relevant references because the paper lacks that to further prove your work's scientific soundness. 

I left a few detailed comments within the manuscript. I hope you find them helpful. 

Regards.

Reviewer 4 Report

Authors should revise the manuscript carefully for spelling mistakes and inappropriate expressions. Overall the work is good and would make a significant contribution to the field of remote sensing. 

Author Response

Point 1: Authors should revise the manuscript carefully for spelling mistakes and inappropriate expressions. Overall the work is good and would make a significant contribution to the field of remote sensing.

Response 1: Thank you for the comments. We have carefully checked the language of the manuscript and submitted it to the professional native English speakers for polish. Please see the latest uploaded revised manuscript.

Round 2

Reviewer 1 Report

My comments have been well revised. 

Reviewer 3 Report

Dear Authors,

Thank you for addressing the proposed suggestions.

Best of luck in your future work.